# *Ex ante* coordination and collusion in zero-sum multi-player extensive-form games

**Gabriele Farina**[*]
Computer Science Department
Carnegie Mellon University
gfarina@cs.cmu.edu

**Andrea Celli**[*]
DEIB
Politecnico di Milano
andrea.celli@polimi.it

**Nicola Gatti**
DEIB
Politecnico di Milano
nicola.gatti@polimi.it

**Tuomas Sandholm**
Computer Science Department
Carnegie Mellon University
sandholm@cs.cmu.edu

## Abstract

Recent milestones in equilibrium computation, such as the success of *Libratus*, show that it is possible to compute strong solutions to two-player zero-sum games in theory and practice. This is not the case for games with more than two players, which remain one of the main open challenges in computational game theory. This paper focuses on zero-sum games where a team of players faces an opponent, as is the case, for example, in Bridge, collusion in poker, and many non-recreational applications such as war, where the colluders do not have time or means of communicating during battle, collusion in bidding, where communication during the auction is illegal, and coordinated swindling in public. The possibility for the team members to communicate before game play—that is, coordinate their strategies *ex ante*—makes the use of behavioral strategies unsatisfactory. The reasons for this are closely related to the fact that the team can be represented as a single player with imperfect recall. We propose a new game representation, the *realization form*, that generalizes the *sequence form* but can also be applied to imperfect-recall games. Then, we use it to derive an *auxiliary game* that is equivalent to the original one. It provides a sound way to map the problem of finding an optimal ex-ante-coordinated strategy for the team to the well-understood Nash equilibrium-finding problem in a (larger) two-player zero-sum perfect-recall game. By reasoning over the auxiliary game, we devise an anytime algorithm, *fictitious team-play*, that is guaranteed to converge to an optimal coordinated strategy for the team against an optimal opponent, and that is dramatically faster than the prior state-of-the-art algorithm for this problem.

## 1   Introduction

In recent years, computational studies on imperfect-information games have largely focused on two-player zero-sum games. In that setting, AI techniques have achieved remarkable results, such as defeating top human specialist professionals in heads-up no-limit Texas hold'em poker [4, 5].

Fewer results are known for settings with more than two players. Yet, many strategic interactions provide players with incentives to team up. In some cases, players may have a similar goal and may be willing to coordinate and share their final reward. Consider, as an illustration, the case of a poker

---

[*]Equal contribution.

game with three or more players, where all but one of them collude against an identified target player and will share the winnings after the game. In other settings, players might be forced to cooperate by the nature of the interaction itself. This is the case, for instance, in the card-playing phase of Bridge, where a team of two players, called the "defenders", plays against a third player, the "declarer". Situations of a team ganging up on a player are, of course, ubiquitous in many non-recreational applications as well, such as war where the colluders do not have time or means of communicating during battle, collusion in bidding where communication during the auction is illegal, coordinated swindling in public, and so on.

The benefits from coordination/collusion depend on the communication possibilities among team members. In this paper, we are interested in *ex ante coordination*, where the team members have an opportunity to discuss and agree on tactics before the game starts, but will be unable to communicate during the game, except through their publicly-observed actions.[2] The team faces an opponent in a zero-sum game (as in, for example, multi-player poker with collusion and Bridge).

Even without communication *during* the game, the planning phase gives the team members an advantage: for instance, the team members could skew their strategies to use certain actions to signal about their state (for example, that they have particular cards). In other words, by having agreed on each member's planned reaction under any possible circumstance of the game, information can be silently propagated in the clear, by simply observing public information.

*Ex ante* coordination can enable the team members to obtain significantly higher utility (up to a factor linear in the number of the game-tree leaves) than the utility they would obtain by abstaining from coordination [1, 6, 25]. Finding an equilibrium with *ex ante* coordination is NP-hard and inapproximable [1, 6]. The only known algorithm is based on a hybrid representation of the game, where team members play joint normal-form actions while the adversary employs sequence-form strategies [6]. We will develop dramatically faster algorithms in this paper.

A team that *ex ante* coordinates can be modeled as a single meta-player. This meta-player typically has imperfect recall, given that the team members observe different aspects of the play (opponent's moves, each others' moves, and chance's moves) and cannot communicate during the game. Then, solving the game amounts to computing a *Nash equilibrium (NE)* in normal-form strategies in a two-player zero-sum imperfect-recall game. The focus on normal-form strategies is crucial. Indeed, it is known that behavioral strategies, that provide a compact representation of the players' strategies, cannot be employed in imperfect-recall games without incurring a loss of expressiveness [18]. Some imperfect-recall games do not even have any NE in behavioral strategies [27]. Even when a NE in behavioral strategies exists, its value can be up to a linear factor (in the number of the game-tree leaves) worse than that of a NE in normal-form strategies. For these reasons, recent efficient techniques for approximating maxmin behavioral strategy profiles in imperfect-recall games [9, 7] are not applicable to our domain.

**Main contributions of this paper.** Our first contribution is a new game representation, which we call the *realization form*. In perfect-recall games it essentially coincides with the sequence form, but, unlike the sequence form, it can also be used in imperfect-recall games. By exploiting the realization form, we produce a two-player *auxiliary game* that has perfect recall, and is equivalent to the normal form of the original game, but significantly more concise. Furthermore, we propose an anytime algorithm, *fictitious team-play*, which is a variation of fictitious play [2]. It is guaranteed to converge to an optimal solution in the setting where the team members coordinate *ex ante*. Experiments show that it is dramatically faster than the prior state-of-the-art algorithm for this problem.

## 2 Preliminaries

In this section we provide a brief overview of extensive-form games (see also the textbook by Shoham and Leyton-Brown [20]). An extensive-form game $\Gamma$ has a finite set $\mathcal{P}$ of players and a finite set of actions $A$. $H$ is the set of all possible nodes, described as sequences of actions (histories). $A(h)$ is the set of actions available at node $h$. If $a \in A(h)$ leads to $h'$, we write $ha = h'$.

$P(h) \in \mathcal{P} \cup \{c\}$ is the player who acts at $h$, where $c$ denotes chance. $H_i$ is the set of decision nodes where player $i$ acts. $Z$ is the set of terminal nodes. For each player $i \in P$, there is a payoff function $u_i : Z \to \mathbb{R}$. An extensive-form game with imperfect information has a set $\mathcal{I}$ of information sets. Decision nodes within the same information set are not distinguishable for the player whose turn it is to move. By definition, for any $I \in \mathcal{I}$, $A(I) = A(h)$, for all $h \in I$. $\mathcal{I}_i$ is the information partition of $H_i$.

A *pure normal-form plan* for player $i$ is a tuple $\sigma \in \Sigma_i = \times_{I \in \mathcal{I}_i} A(I)$ that specifies an action for each information set of that player. $\sigma(I)$ denotes the action selected in $\sigma$ at information set $I$. A *normal-form strategy* $x_i$ for player $i$ is defined as $x_i : \Sigma_i \to \Delta^{|\Sigma_i|}$. We denote by $\mathcal{X}_i$ the normal-form strategy space of player $i$. A *behavioral strategy* $\pi_i \in \Pi_i$ associates each $I \in \mathcal{I}_i$ with a probability vector over $A(I)$. $\pi_i(I, a)$ denotes the probability with which $i$ chooses action $a$ at $I$. $\pi_c$ is the strategy of a virtual player, "chance", who plays non-strategically and is used to represent exogenous stochasticity. The expected payoff of player $i$, when she plays $x_i$ and the opponents play $x_{-i}$, is denoted, with an overload of notation, by $u_i(x_i, x_{-i})$.

Denote by $\rho_i^{x_i}(z)$ the probability with which player $i$ plays to reach $z$ when following strategy $x_i$ ($\rho_i^{\pi_i}(z)$ is defined analogously). Then, $\rho^x(z) = \prod_{i \in P \cup \{c\}} \rho_i^{x_i}(z)$ is the probability of reaching $z$ when players follow behavioral strategy profile $x$. We say that $x_i, x_i'$ are *realization equivalent* if, for any $x_{-i}$ and for any $z \in Z$, $\rho^x(z) = \rho^{x'}(z)$, where $x = (x_i, x_{-i})$, $x' = (x_i', x_{-i})$. The same definition holds for strategies in different representations (e.g., behavioral and sequence form). Similarly, two strategies $x_i, x_i'$ are *payoff equivalent* if, $\forall j \in \mathcal{P}$ and $\forall x_{-i}$, $u_j(x_i, x_{-i}) = u_j(x_i', x_{-i})$.

A player has *perfect recall* if she has perfect memory of her past actions and observations. Formally, $\forall x_i, \forall I \in \mathcal{I}_i, \forall h, h' \in I, \rho_i^{x_i}(h) = \rho_i^{x_i}(h')$. $\Gamma$ has perfect recall if every player has perfect recall.

$\mathsf{BR}(x_{-i})$ denotes the *best response* of player $i$ against a strategy profile $x_{-i}$. A best response is a strategy such that $u_i(\mathsf{BR}(x_{-i}), x_{-i}) = \max_{x_i \in \mathcal{X}_i} u_i(x_i, x_{-i})$. A NE [17] is a strategy profile in which no player can improve her utility by unilaterally deviating from her strategy. Therefore, for each player $i$, a NE $x^* = (x_i^*, x_{-i}^*)$ satisfies $u_i(x_i^*, x_{-i}^*) = u_i(\mathsf{BR}(x_{-i}^*), x_{-i}^*)$.

The sequence form [12, 23] of a game is a compact representation applicable only to games with perfect recall. It decomposes strategies into sequences of actions and their realization probabilities. A sequence $q_i \in Q_i$ for player $i$, defined by a node $h$, is a tuple specifying player $i$'s actions on the path from the root to $h$. A sequence is said terminal if, together with some sequences of the other players, leads to a terminal node. $q_\emptyset$ denotes the fictitious sequence leading to the root node and $qa$ is the extended sequence obtained by appending action $a$ to $q$. A *sequence-form strategy* for player $i$ is a function $r_i : Q_i \to [0, 1]$, s.t. $r_i(q_\emptyset) = 1$ and, for each $I \in \mathcal{I}_i$ and sequence $q$ leading to $I$, $-r_i(q) + \sum_{a \in A(I)} r_i(qa) = 0$.

## 3 Team-maxmin equilibrium with coordination device (TMECor)

In the setting of *ex ante coordination*, team members have the opportunity to discuss tactics before the game begins, but are otherwise unable to communicate during the game, except via publicly-observed actions. A powerful, game-theoretic way to think about *ex ante* coordination is through a *coordination device*. In the planning phase before the game starts, the team members identify a set of joint pure normal-form plans. Then, just before the play, the coordination device will randomly draw one of the normal-form plans from a given probability distribution, and the team members will all act as specified in the selected plan. A NE where team members play *ex ante* coordinated normal-form strategies is called a *team-maxmin equilibrium with coordination device* (TMECor) [6].[3] In an approximate version, $\epsilon$-TMECor, neither the team nor the opponent can gain more than $\epsilon$ by deviating from their strategy, assuming that the other does not deviate.

By sampling a recommendation from a joint probability distribution over $\Sigma_1, \Sigma_2$, the coordination device introduces a correlation between the strategies of the team members that is otherwise impossible to capture using behavioral strategies. In other words, in general there exists no behavioral strategy for the team player that is realization-equivalent to the normal-form strategy induced by the coordination device, as the following example further illustrates.

**Example 1.** *Consider the zero-sum game in Figure 1. Two team members (Players 1 and 2) play against an adversary $\mathcal{A}$. The team obtains a cumulative payoff of 2 when the game ends at ① or ⑧, and a payoff of 0 otherwise. A valid* ex ante *coordination device is as follows: the team members toss an unbiased coin; if heads comes up, Player 1 will play action A and Player 2 will play action C; otherwise, Player 1 will play action B and Player 2 will play action D. The realization induced on the leaves is such that $\rho(①) = \rho(⑧) = 1/2$ and $\rho(ⓘ) = 0$ for $i \notin \{1,8\}$. No behavioral strategy for the team members is able to induce the same realization. This coordination device is enough to overcome the imperfect information of Player 2 about Player 1's move, as Player 2 knows what action will be played by Player 1 even though Player 2 will not observe it during the game.*

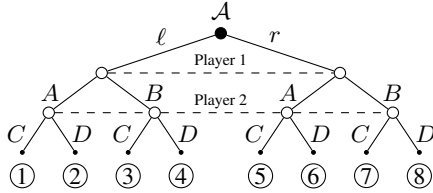

Figure 1: Example of extensive-form game with a team. The uppercase letters denote the action names. The circled numbers uniquely identify the terminal nodes.

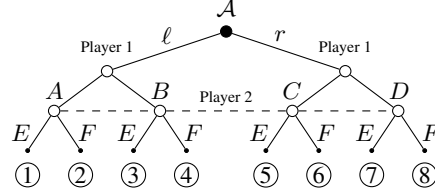

Figure 2: A game where coordinated strategies have a weak signaling power. The uppercase letters denote the action names. The circled numbers uniquely identify the terminal nodes.

One might wonder whether there is value in forcing the coordination device to only induce normal-form strategies for which a realization-equivalent tuple of behavioral strategies (one for each team member) exists. Indeed, under such a restriction, the problem of constructing an optimal coordination device would amount to finding the optimal tuple of behavioral strategies (one for each team member) that maximizes the team's utility. This solution concept is known as *team-maxmin equilibrium (TME)* [25]. TME offers conceptual simplicity that unfortunately comes at a high cost. First, finding the best tuple of behavioral strategies is a non-linear, non-convex optimization problem. Moreover, restricting to TMEs is also undesirable in terms of final utility for the team, since it may incur in an arbitrarily large loss compared to a TMECor [6].

Interestingly, as we will prove in Section 4.2, there is a strong connection between TME and TMECor. The latter solution concept can be seen as the natural "convexification" of the former, in a sense that we will make precise in Theorem 2.

## 4   *Realization form:* **a universal, low-dimensional game representation**

In this section, we introduce the *realization form* of a game, which enables one to represent the strategy space of a player by a number of variables that is linear in the game size (as opposed to exponential as in the normal form), even in games with imperfect recall. For each player $i$, a *realization-form strategy* is a vector that specifies the probability with which $i$ plays to reach the different terminal nodes. The mapping from normal-form strategies to realization-form strategies allows us to compress the action space from $\mathcal{X}_i$, which has as many coordinates as the number of normal-form plans—usually exponential in the size of the tree—to a space that has one coordinate for each terminal node. This mapping is many-to-one because of the redundancies in the normal-form representation. Given a realization-form strategy, all the normal-form strategies that induce it are payoff equivalent.

The construction of the realization form relies on the following observation.

**Observation 1.** *Let $\Gamma$ be a game and $z \in Z$ be a terminal node. Given a normal-form strategy profile $x = (x_1, \dots, x_n) \in \mathcal{X}_1 \times \cdots \times \mathcal{X}_n$, the probability of reaching $z$ can be uniquely decomposed as the product of the contributions of each individual player, plus chance's contribution. Formally, $\rho^x(z) = \rho_c^{x_c} \prod_{i \in \mathcal{P}} \rho_i^{x_i}(z)$.*

**Definition 1** (Realization function)**.** *Let $\Gamma$ be a game. The* realization function *of player $i \in \mathcal{P}$ is the function $f_i^\Gamma : \mathcal{X}_i \to [0,1]^{|Z|}$ that maps every normal-form strategy for player $i$ to the corresponding vector of realizations for each terminal node: $f_i^\Gamma : \mathcal{X}_i \ni x \mapsto (\rho_i^x(z_1), \dots, \rho_i^x(z_{|Z|}))$.*

We are interested in the range of $f_i^\Gamma$, called the *realization polytope* of player $i$.

**Definition 2** (Realization polytope and strategies). *Player $i$'s realization polytope $\Omega_i^\Gamma$ in game $\Gamma$ is the range of $f_i^\Gamma$, that is the set of all possible realization vectors for player $i$: $\Omega_i^\Gamma := f_i^\Gamma(\mathcal{X}_i)$. We call an element $\omega_i \in \Omega_i^\Gamma$ a* realization-form strategy *(or, simply, realization) of player $i$.*

The function that maps a tuple of realization-form strategies, one for each player, to the payoff of each player, is multilinear. This is by construction and follows from Observation 1. Moreover, the realization function has the following strong property (all proofs are provided in Appendix B).

**Lemma 1.** $f_i^\Gamma$ *is a linear function and $\Omega_i^\Gamma$ is a convex polytope.*

For players with perfect recall, the realization form is the projection of the sequence form, where variables related to non-terminal sequences are dropped. In other words, when the perfect-recall property is satisfied, it is possible to move between the sequence-form and the realization-form representations by means of a simple linear transformation. Therefore, the realization polytope of perfect-recall games can be described with a linear number (in the game size) of linear constraints. Conversely, in games with imperfect recall the number of constraints required to describe the realization polytope may be exponential[4]. A key feature of the realization form is that it can be applied to both settings without any modification. For example, an optimal NE in a two-player zero-sum game, with or without perfect recall and/or information, can be computed through the bilinear saddle-point problem $\max_{\omega_1 \in \Omega_1^\Gamma} \min_{\omega_2 \in \Omega_1^\Gamma} \omega_1^\top U \omega_2$, where $U$ is a (diagonal) $|Z| \times |Z|$ *payoff* matrix.

Finally, the realization form of a game is formally defined as follows.

**Definition 3** (Realization form). *Given an extensive-form game $\Gamma$, its realization form is a tuple $(\mathcal{P}, Z, U, \Omega^\Gamma)$, where $\Omega^\Gamma$ specifies a realization polytope for each $i \in \mathcal{P}$.*

### 4.1 Two examples of realization polytopes

To illustrate the realization-form construction, we consider two three-player zero-sum extensive-form games with perfect recall, where a team composed of two players playing against the third player. As already observed, since the team member have the same incentives, the team as a whole behaves as a single meta-player with (potentially) imperfect recall. As we show in Example 2, ex-ante coordination allows team members to behave as a single player with perfect recall. In contrast, in Example 3, the signaling power of *ex ante* coordinated strategies is not enough to fully reveal private team members' information.

**Example 2.** *Consider the game depicted in Figure 1. $\mathcal{X}_\mathcal{T}$ is the 4-dimensional simplex corresponding to the space of probability distributions over the set of pure normal-form plans $\Sigma_\mathcal{T} = \{AC, AD, BC, BD\}$. Given $x \in \mathcal{X}_\mathcal{T}$, the probability with which $\mathcal{T}$ plays to reach a certain outcome is the sum of every $x(\sigma)$ such that plan $\sigma \in \Sigma_\mathcal{T}$ is consistent with the outcome (i.e., the outcome is reachable if $\mathcal{T}$ plays $\sigma$). In the example, we have:*

$$f_\mathcal{T}(x) = (x(A,C),\ x(A,D),\ x(B,C),\ x(B,D),\ x(A,C),\ x(A,D),\ x(B,C),\ x(B,D)),$$

*where outcomes are ordered from left to right in the tree. Then, the realization polytope is described by Polytope 1. These constraints show that Player $\mathcal{T}$ has perfect recall when employing coordinated strategies. Indeed, the constraints coincides with the sequence-form constraints obtained when splitting Player 2's information set into two information sets, one for each action $\{A, B\}$.*

**Example 3.** *In the game in Figure 2, the team Player $\mathcal{T}$ has imperfect recall even when coordination is allowed. In this case, the signaling power of* ex ante *coordinated strategies is not enough for Player 1 to propagate the information observed (that is, $\mathcal{A}$'s move) to Player 2. It can be verified that the realization polytope $\Omega_\mathcal{T}^\Gamma$ is characterized by the set of constraints in Polytope 2 (see Appendix A for more details). As one might expect, this polytope contains Polytope 1.*

### 4.2 Relationship with team max-min equilibrium

In this subsection we study the relationship with team max-min equilibrium, and prove a fact of potential independent interest. This subsection is not needed for understanding the rest of the paper.

We prove that the realization polytope of a non-absent-minded player is the convex hull of the set of realizations that are reachable starting from behavioral strategies. This gives a precise meaning to our claim that *the TMECor concept is the convexification of the TME concept.*

$$\begin{cases} \omega(\text{⑤}) + \omega(\text{⑥}) + \omega(\text{⑦}) + \omega(\text{⑧}) = 1, \\ \omega(\text{②}) = \omega(\text{⑥}), \quad \omega(\text{④}) = \omega(\text{⑧}), \\ \omega(\text{①}) = \omega(\text{⑤}), \quad \omega(\text{③}) = \omega(\text{⑦}), \\ \omega(\text{⑦}) \geq 0 \quad i \in \{1,2,3,4,5,6,7,8\}. \end{cases}$$

Polytope 1: Description of the realization polytope for the game of Figure 1.

$$\begin{cases} \omega(\text{⑤}) + \omega(\text{⑥}) + \omega(\text{⑦}) + \omega(\text{⑧}) = 1, \\ \omega(\text{②}) + \omega(\text{④}) = \omega(\text{⑥}) + \omega(\text{⑧}), \\ \omega(\text{①}) + \omega(\text{③}) = \omega(\text{⑤}) + \omega(\text{⑦}), \\ \omega(\text{⑦}) \geq 0 \quad i \in \{1,2,3,4,5,6,7,8\}. \end{cases}$$

Polytope 2: Description of the realization polytope for the game of Figure 2.

**Definition 4.** *Let* $\Gamma$ *be a game. The* behavioral-realization function *of player* $i$ *is the function* $\tilde{f}_i^\Gamma :$ $\Pi_i \ni \pi \mapsto (\rho_i^\pi(z_1), \ldots, \rho_i^\pi(z_{|Z|})) \in [0,1]^{|Z|}$. *Accordingly, the* behavioral-realization set *of player* $i$ *is the range of* $\tilde{f}_i^\Gamma$, *that is* $\tilde{\Omega}_i^\Gamma := \tilde{f}_i^\Gamma(\Pi_i)$. *This set is generally non-convex.*

Denoting by $\text{co}(\cdot)$ the convex hull of a set, we have the following:

**Theorem 2.** *Consider a game* $\Gamma$. *If player* $i$ *is not absent-minded, then* $\Omega_i^\Gamma = \text{co}(\tilde{\Omega}_i^\Gamma)$.

## 5  *Auxiliary game:* an equivalent game that enables the use of behavioral strategies

In the rest of this paper, we focus on three-player zero-sum extensive-form games with perfect recall, and we will model the interaction of a team composed of two players playing against the third player. The theory developed also applies to settings with teams with an arbitrary number of players.

We prove that it is possible to construct an *auxiliary game* with the following properties:

- it is a two-player perfect-recall game between the adversary $\mathcal{A}$ and a *team-player* $\mathcal{T}$;
- for both players, the set of behavioral strategies is as "expressive" as the set of the normal-form strategies in the original game (i.e., in the case of the team, the set of strategies that team members can achieve through *ex ante* coordination).

To accomplish this, we introduce a root node $\phi$, whose branches correspond to the normal-form strategies of the first player of the team. This representation enables the team to express any probability distribution over the ensuing subtrees, and leads to an equivalence between the behavioral strategies in this new perfect-recall game (the auxiliary game) and the normal-form strategies of the original two-player imperfect-recall game between the team and the opponent. The auxiliary game is a perfect-recall representation of the original imperfect-recall game such that the expressiveness of behavioral (and sequence-form) strategies is increased to match the expressiveness of normal-form strategies in the original game.

Consider a generic $\Gamma$ with $\mathcal{P} = \{1, 2, \mathcal{A}\}$, where 1 and 2 are team members. We will refer to Player 1 as the *pivot player*. For any $\sigma_1 \in \Sigma_1$, we define $\Gamma_{\sigma_1}$ as the two-player game with $\mathcal{P} = \{2, \mathcal{A}\}$ that we obtain from $\Gamma$ by fixing the choices of Player 1 as follows: $\forall I \in \mathcal{I}_1$ and $\forall a \in A(I)$, if $a = \sigma_1(I)$, then $\pi_{1,\sigma_1}(I, a) = 1$; otherwise, $\pi_{1,\sigma_1}(I, a) = 0$. Once $\pi_{1,\sigma_1}$ has been fixed in $\Gamma_{\sigma_1}$, decision nodes belonging to Player 1 can be considered as if they were chance nodes. The auxiliary game of $\Gamma$, denoted with $\Gamma^*$, is defined as follows.

**Definition 5** (Auxiliary Game)**.** *The auxiliary game* $\Gamma^*$ *is a two-player game obtained from* $\Gamma$ *in the following way:*

- $\mathcal{P} = \{\mathcal{T}, \mathcal{A}\}$;
- *the root* $\phi$ *is a decision node of Player* $\mathcal{T}$ *with* $A(\phi) = \{a_\sigma\}_{\sigma \in \Sigma_1}$;
- *each* $a_\sigma$ *is followed by a subtree* $\Gamma_\sigma$;
- $\mathcal{A}$ *does not observe the action chosen by* $\mathcal{T}$ *at* $\phi$.

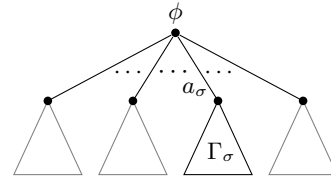

Figure 3: Structure of the auxiliary game $\Gamma^*$.

By construction, all the decision nodes of any information set of team $\mathcal{T}$ are part of the same subtree $\Gamma_\sigma$. Intuitively, this is because, in the original game, team members jointly pick an action from their joint probability distribution and, therefore, every team member knows what the other member is going to play. The opponent has the same number of information sets both in $\Gamma$ and $\Gamma^*$. This is because she does not observe the choice at $\phi$ and, therefore, her information sets span across all subtrees $\Gamma_\sigma$. The basic structure of the auxiliary game tree is depicted in Figure 3 (information sets of $\mathcal{A}$ are omitted for clarity). Games with more than two team members can be represented though

a $\Gamma^*$ which has a number of subtrees equal to the Cartesian product of the normal-form plans of all team members except one.

The next lemma is fundamental to understand the equivalence between behavioral strategies of $\Gamma^*$ and normal-form strategies of $\Gamma$. Intuitively, it justifies the introduction of the root node $\phi$, whose branches correspond to the normal-form strategies of the pivot player. This representation enables the team $\mathcal{T}$ to express any convex combination of realizations in the $\Gamma_\sigma$ subtrees.

**Lemma 3.** *For any* $\Gamma$, $\Omega_{\mathcal{T}}^\Gamma = \mathrm{co}\left(\bigcup_{\sigma \in \Sigma_1} \Omega_{\mathcal{T}}^{\Gamma_\sigma}\right)$.

The following theorem follows from Lemma 3 and characterizes the relationship between $\Gamma$ and $\Gamma^*$. It shows that there is a strong connection between the strategies of Player $\mathcal{T}$ in the auxiliary game and the *ex ante* coordinated strategies for the team members in the original game $\Gamma$.

**Theorem 4.** *Games* $\Gamma$ *and* $\Gamma^*$ *are realization-form equivalent in the following sense:*

    *(i)* ***Team****. Given any distribution over the actions at the game tree root* $\phi$ *(i.e., a choice* $\Sigma_1 \ni \sigma \mapsto \lambda_\sigma \geq 0$ *such that* $\sum_\sigma \lambda_\sigma = 1$*) and any choice of realizations* $\{\omega_\sigma \in \Omega_{\mathcal{T}}^{\Gamma_\sigma}\}_{\sigma \in \Sigma_1}$*, we have that* $\sum_{\sigma \in \Sigma_1} \lambda_\sigma \omega_\sigma \in \Omega_{\mathcal{T}}^\Gamma$*. The converse is also true: given any* $\omega \in \Omega_{\mathcal{T}}^\Gamma$*, there exists a choice of* $\{\lambda_\sigma\}_{\sigma \in \Sigma_1}$ *and realizations* $\{\omega_\sigma \in \Omega_{\mathcal{T}}^{\Gamma_\sigma}\}_{\sigma \in \Sigma_1}$ *such that* $\omega = \sum_{\sigma \in \Sigma_1} \lambda_\sigma \omega_\sigma$*.*

    *(ii)* ***Adversary****. The realization polytope of the adversary satisfies* $\Omega_{\mathcal{A}}^\Gamma = \Omega_{\mathcal{A}}^{\Gamma^*}$*.*

The following is then a direct consequence of Theorem 4

**Corollary 1.** *The set of payoffs reachable in* $\Gamma$ *coincides with the set of payoffs reachable in* $\Gamma^*$*. Specifically, any strategy* $\{\lambda_\sigma\}_{\sigma \in \Sigma_1}$*,* $\{\omega_\sigma\}_{\sigma \in \Sigma_1}$ *over* $\Gamma^*$ *is payoff-equivalent to the realization-form strategy* $\omega = \sum_{\sigma \in \Sigma_1} \lambda_\sigma \omega_\sigma$ *in* $\Gamma$*.*

**Remark 1.** *Since* $\Gamma_\sigma$ *has perfect recall, every realization* $\omega_\sigma \in \Omega^{\Gamma_\sigma}$ *can be induced by* $\mathcal{T}$ *via behavioral strategies.*

The above shows that for every *ex ante* coordinated strategy for the team in $\Gamma$, there exists a corresponding (payoff-equivalent) behavioral strategy for $\mathcal{T}$ in $\Gamma^*$, and *vice versa*. Hence, due to realization-form equivalence between $\Gamma$ and $\Gamma^*$, finding a TMECor in $\Gamma$ (employing *ex ante* coordinated normal-form strategies), is equivalent to finding a NE in $\Gamma^*$ (with behavioral strategies).

## 6   *Fictitious team-play:* an anytime algorithm for TMECor

This section introduces an anytime algorithm, *fictitious team-play*, for finding a TMECor. It follows from the previous section that in order to find a TMECor in $\Gamma$, it suffices to find a two-player NE in the auxiliary game $\Gamma^*$ (and vice versa, although we do not use this second direction). Furthermore, since $\Gamma^*$ is a two-player perfect-recall zero-sum game, the *fictitious play (FP)* algorithm can be applied with its theoretical guarantee of converging to a NE. Fictitious play [2, 19] is an iterative algorithm originally described for normal-form games. It keeps track of average normal-form strategies $\bar{x}_i$, which are output in the end, and they converge to a NE. At iteration $t$, player $i$ computes the best response against the opponent's empirical distribution of play up to time $t - 1$, that is, $x_i^t = \mathsf{BR}(\bar{x}_{-i}^{t-1})$. Then her average strategy is updated as $\bar{x}_i^t = \frac{t-1}{t}\bar{x}_i^{t-1} + \frac{1}{t}x_i^t$. Conceptually, our fictitious team-play algorithm coincides with FP applied to the auxiliary game $\Gamma^*$. However, in order to avoid the exponential size of $\Gamma^*$, our fictitious team-play algorithm does not explicitly work on the auxiliary game. Rather, it encodes the best-response problems by means of mixed integer linear programs (MILPs) on the original game $\Gamma$.

**The main algorithm**. The pseudocode of the main algorithm is given as Algorithm 1, where $\mathsf{BR}_{\mathcal{A}}(\cdot)$ and $\mathsf{BR}_{\mathcal{T}}(\cdot)$ are the subroutines for solving the best-response problems.

Our algorithm employs realization-form strategies. This allows for a significantly more intuitive way of performing averaging (Steps 7, 8, 10) than what is done in full-width extensive-form fictitious play [11], which employs behavioral strategies.

Our algorithm maintains an average realization $\bar{\omega}_{\mathcal{A}}$ for the adversary. Moreover, the $|\Sigma_1|$-dimensional vector $\bar{\lambda}$ keeps the empirical frequencies of actions at node $\phi$ in auxiliary game $\Gamma^*$ (see Figure 3). Finally, $\forall \sigma \in \Sigma_1, \bar{\omega}_{\mathcal{T},\sigma} \in \Omega_{\mathcal{T}}^{\Gamma_\sigma}$ is the average realization of the team in the subtree $\Gamma_\sigma$.

After $t$ iterations of the algorithm, only $t$ pairs of strategies are generated. Hence, an optimized implementation of the algorithm can employ a lazy data structure to keep track of the changes to $\bar{\lambda}$ and $\bar{\omega}_{\mathcal{T},\sigma}$.

Initially (Step 2), the average realization $\bar{\omega}_{\mathcal{A}}$ of the adversary is set to the realization-form strategy equivalent to a uniform behavioral strategy profile. At each iteration the algorithm first computes a team's best-response against $\bar{\omega}_{\mathcal{A}}$. We require that the chosen best response assign probability 1 to one of the available actions (say, $a_{\sigma^t}$) at node $\phi$. (A pure—that is, non-randomized—best response always exists and, therefore, in particular there always exists

---

**Algorithm 1** Fictitious team-play

1: **function** FICTITIOUSTEAMPLAY($\Gamma$)
2:      Initialize $\bar{\omega}_{\mathcal{A}}$
3:      $\bar{\lambda} \leftarrow (0, \ldots, 0), t \leftarrow 1$
4:      $\bar{\omega}_{\mathcal{T},\sigma} \leftarrow (0, \ldots, 0) \quad \forall \sigma \in \Sigma_1$
5:      **while** within computational budget **do**
6:          $(\sigma^t, \omega_{\mathcal{T}}^t) \leftarrow \mathsf{BR}_{\mathcal{T}}(\bar{\omega}_{\mathcal{A}})$
7:          $\bar{\lambda} \leftarrow (1 - \frac{1}{t})\bar{\lambda} + \frac{1}{t}\mathbb{1}_{\sigma^t}$
8:          $\bar{\omega}_{\mathcal{T},\sigma^t} \leftarrow (1 - \frac{1}{t})\bar{\omega}_{\mathcal{T},\sigma^t} + \frac{1}{t}\omega_{\mathcal{T}}^t$
9:          $\omega_{\mathcal{A}}^t \leftarrow \mathsf{BR}_{\mathcal{A}}(\bar{\lambda}, \{\bar{\omega}_{\mathcal{T},\sigma}\}_\sigma)$
10:         $\bar{\omega}_{\mathcal{A}} \leftarrow (1 - \frac{1}{t})\bar{\omega}_{\mathcal{A}} + \frac{1}{t}\omega_{\mathcal{A}}^t$
11:         $t \leftarrow t + 1$
12:      **return** $(\bar{\lambda}, (\bar{\omega}_{\mathcal{T},\sigma})_{\sigma \in \Sigma_1})$

---

at least one best response selecting a single action at the root with probability one.) Then, the average frequencies and team's realizations are updated on the basis of the observed $(\sigma^t, \omega_{\mathcal{T}}^t)$. Finally, the adversary's best response $\omega_{\mathcal{A}}^t$ against the updated average strategy of the team is computed, and the empirical distribution of play of the adversary is updated.

The *ex ante* coordinated strategy profile for the team is implicitly represented by the pair $(\bar{\lambda}, \bar{\omega}_{\mathcal{T},\sigma})$. In particular, that pair encodes a coordination device that operates as follows:

- At the beginning of the game, a pure normal-form plan $\tilde{\sigma} \in \Sigma$ is sampled according to the discrete probability distribution encoded by $\bar{\lambda}$. Player 1 will play the game according to the sampled plan.
- Player 2 will play according to any normal-form strategy in $f_2^{-1}(\bar{\omega}_{\mathcal{T},\tilde{\sigma}})$, that is, any normal-form strategy whose realization is $\bar{\omega}_{\mathcal{T},\tilde{\sigma}}$.

The correctness of the algorithm then is a direct consequence of realization-equivalence between $\Gamma$ and $\Gamma^*$, which was shown in Theorem 4. In particular, the strategy of the team converges to a profile that is part of a normal-form NE in the original game $\Gamma$.

**Best-response subroutines**. The problem of finding the adversary's best response to a pair of strategies of the team, namely $\mathsf{BR}_{\mathcal{A}}(\bar{\lambda}, \{\bar{\omega}_{\mathcal{T},\sigma}\}_\sigma)$, can be efficiently tackled by working on $\Gamma$ (second point of Theorem 4). In contrast, the problem of computing $\mathsf{BR}_{\mathcal{T}}(\bar{\omega}_{\mathcal{A}})$ is NP-hard [24], and inapproximable [6]. Celli and Gatti [6] propose a MILP formulation to solve the team best-response problem. In Appendix E we propose an alternative MILP formulation in which the number of binary variables is polynomial in $\Gamma$ and proportional to the number of sequences of the *pivot* player.

In our algorithm, we employ a *meta-oracle* that uses simultaneously, as parallel processes, both subroutines, and stops them as soon as one of the two has found a solution or, in the case a time-limit is reached, it stops both subroutines and it returns the best solution (in terms of team's utility). This circumvents the need to prove optimality in the MILP, which often takes most of the MILP-solving time, and opens the doors to heuristic MILP-solving techniques. One of the key features of the meta-oracle is that its performances are not impacted by the size of $\Gamma^*$, which is never *explicitly* employed in the best-responses computation.

## 7 Experiments

We conducted experiments on three-player Kuhn poker games and three-player Leduc hold'em poker games. These are standard games in the computational game theory literature, and description of them can be found in Appendix F. Our instances are parametric in the number of ranks in the deck. The instances adopted are listed in Tables 1 and 2, where K$r$ and L$r$ denote, respectively, a Kuhn instance with $r$ ranks and a Leduc instance with $r$ ranks (i.e., $3r$ total cards). Table 1 also displays the instances' dimensions in terms of the number of information sets per player and the number of sequences (i.e., number of information set–action pairs) per player, as well as the payoff dispersion $\Delta_u$—that is, the difference between the maximum and minimum attainable team utility.

**Fictitious team-play.** We instantiated fictitious team-play with the meta-oracle previously discussed, which returns the best solution found by the MILP oracles within the time limit. We let each best-response formulation run on the Gurobi 8.0 MILP solver, with a time limit of 15 seconds and 5000 maximum iterations. Our algorithm is an anytime algorithm, so it does not require a target accuracy $\epsilon$ for $\epsilon$-TMECor to be specified in advance. Table 1 shows the any-

| Game | Tree size | | $\Delta_u$ | Fictitious team-play | | | | | | HCG |
|---|---|---|---|---|---|---|---|---|---|---|
| | Inf. | Seq. | | 10% | 5% | 2% | 1.5% | 1% | 0.5% | |
| K3 | 25 | 13 | 6 | 0s | 0s | 0s | 1s | 1s | 1s | 0s |
| K4 | 33 | 17 | 6 | 1s | 1s | 4s | 4s | 30s | 1m 12s | 9s |
| K5 | 41 | 21 | 6 | 1s | 2s | 44s | 1m | 4m 15s | 8m 57s | 1m 58s |
| K6 | 49 | 25 | 6 | 1s | 12s | 43s | 5m 15s | 8m 30s | 23m 32s | 25m 26s |
| K7 | 57 | 29 | 6 | 4s | 17s | 2m 15s | 5m 46s | 6m 31s | 23m 49s | 2h 50m |
| L3 | 457 | 229 | 21 | 15s | 1m | 14m 05s | 30m 40s | 1h 34m 30s | > 24h | oom |
| L4 | 801 | 401 | 21 | 1s | 1m 31s | 11m 8s | 51m 5s | 6h 51m | > 24h | oom |

Table 1: Comparison between the run times of fictitious team-play (for various levels of accuracy) and the hybrid column generation (HCG) algorithm. (*oom*: out of memory.)

| Game | Team Utility | | |
|---|---|---|---|
| | Adv 1 | Adv 2 | Adv 3 |
| K3 | 0.0000 | 0.0000 | 0.0003 |
| K4 | 0.0405 | 0.0259 | -0.0446 |
| K5 | 0.0434 | 0.0156 | -0.0282 |
| K6 | 0.0514 | 0.0271 | -0.0253 |
| K7 | 0.0592 | 0.0285 | -0.0259 |
| L3 | 0.2332 | 0.2089 | 0.1475 |
| L4 | 0.1991 | 0.1419 | -0.0223 |

Table 2: Values of the average strategy profile for different choices of adversary.

time performance, that is, the time it took to reach an $\alpha\Delta_u$-TMECor for different accuracies $\alpha \in \{10\%, 5\%, 2\%, 1.5\%, 1\%, 0.5\%\}$. Results in Table 1 assume that the team consists of the first and third mover in the game; the opponent is the second mover. Table 2 shows the value of the average strategy computed by fictitious team-play for different choices of the opponent player. This value corresponds to the expected utility of the team for the average strategy profile $(\bar{\lambda}, \bar{\omega}_{\mathcal{T},\sigma})$ at iteration 1000. In Appendix F.3 we show the minimum cumulative utility that the team is *guaranteed* to achieve, that is $-\mathsf{BR}_{\mathcal{A}}(\bar{\lambda}, \{\bar{\omega}_{\mathcal{T},\sigma}\}_\sigma)$.

**Hybrid column generation benchmark**. We compared against the hybrid column generation (HCG) algorithm [6], which is the only prior algorithm for this problem. To make the comparison fair, we instantiate HCG with the same meta-oracle discussed in the previous section. We again use Gurobi 8.0 MILP solver to solve the best response problem for the team. However, in the case of HCG, no time limit can be set on Gurobi without invalidating the theoretical convergence guarantee of the algorithm. This is a drawback, as it prevents HCG from running in an *anytime* fashion, despite column generation otherwise being an anytime algorithm. In the Leduc poker instances, HCG exceeded the memory budget (40 GB).

Our experiments show that fictitious team-play scales to significantly larger games than HCG. Interestingly, in almost all the games, the value of the team was non-negative: by colluding, the team was able to achieve victory. Moreover, in Appendix F.4, we show that a TMECor provides to the team a substantial payoff increase over the setting where team members play in behavioral strategies.

## 8 Conclusions and future research

The study of algorithms for multi-player games is challenging. In this paper, we proposed an algorithm for settings in which a team of players faces an adversary and the team members can exploit only *ex ante* coordination, discussing and agreeing on tactics before the game starts. Our first contribution was the realization form, a novel representation that allows us to represent the strategies of the normal form more concisely. The realization form is also applicable to imperfect-recall games. We used it to derive a two-player perfect-recall auxiliary game that is equivalent to the original game, and provides a theoretically sound way to map the problem of finding an optimal ex-ante-coordinated strategy for the team to a classical well-understood Nash equilibrium-finding problem in a two-player zero-sum perfect-recall game. Our second contribution was the design of the fictitious team-play algorithm, which employs a novel best-response meta-oracle. The anytime algorithm is guaranteed to converge to an equilibrium. Our experiments showed that fictitious team-play is dramatically faster than the prior algorithms for this problem.

In the future, it would be interesting to adapt other popular equilibrium computation techniques from the two-player setting (such as CFR) for our setting, by reasoning over the auxiliary game.

The study of algorithms for team games could shed further light on how to deal with imperfect-recall games, that are receiving increasing attention in the community due to the application of imperfect-recall abstractions to the computation of strategies for large extensive-form games [26, 14, 10, 3, 8, 13].

**Acknowledgments.** This material is based on work supported by the National Science Foundation under grants IIS-1718457, IIS-1617590, and CCF-1733556, and the ARO under award W911NF-17-1-0082.

## Footnotes

[2]This kind of coordination has sometimes been referred to as *ex ante correlation* among team members [6]. However, we will not use that term because this setting is quite different than the usual notion of correlation in game theory. In the usual correlation setting, the individual players have to be incentivized to follow the recommendations of the correlation device. In contrast, here there is no need for such incentives because the members of the team can share the benefits from coordination.

[3]They actually called it correlation, not coordination. As explained in the introduction, we will use the term coordination. However, we will keep their acronym TMECor instead of switching to the acronym TMECoor.

[4]Understanding in which subclasses of imperfect-recall games the number of constraints remains polynomial is an interesting open problem.

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
