[Supplementary Material]

# A  Example 3 (continued)

Denote by $\Gamma$ the game depicted in Figure 2. Table 3 shows the value of the realization function, evaluated in each pure normal-form plans of Player $\mathcal{T}$. Each row is a realization vector, and the realization polytope $\Omega_{\mathcal{T}}^{\Gamma}$ is the convex hull of all these vectors.

|       | ① | ② | ③ | ④ | ⑤ | ⑥ | ⑦ | ⑧ |
|-------|---|---|---|---|---|---|---|---|
| $ACE$ | 1 | 0 | 0 | 0 | 1 | 0 | 0 | 0 |
| $ACF$ | 0 | 1 | 0 | 0 | 0 | 1 | 0 | 0 |
| $ADE$ | 1 | 0 | 0 | 0 | 0 | 0 | 1 | 0 |
| $ADF$ | 0 | 1 | 0 | 0 | 0 | 0 | 0 | 1 |
| $BCE$ | 0 | 0 | 1 | 0 | 1 | 0 | 0 | 0 |
| $BCF$ | 0 | 0 | 0 | 1 | 0 | 1 | 0 | 0 |
| $BDE$ | 0 | 0 | 1 | 0 | 0 | 0 | 1 | 0 |
| $BDF$ | 0 | 0 | 0 | 1 | 0 | 0 | 0 | 1 |

Table 3: Mapping between pure normal-form plans and their images under the realization function.

# B  Proofs

**Lemma 1.** *$f_i^{\Gamma}$ is a linear function and $\Omega_i^{\Gamma}$ is a convex polytope.*

*Proof.* We start by proving that $f_i$ is linear. Fix a terminal node $z \in Z$, and define $\Sigma_i^*(z)$ as the subset of pure normal-form plans $\Sigma_i$ of player $i$ for which there exists at least a choice of pure normal-form plans, one for each of the other players, such that under that choice the game terminates in $z$. Given a normal-form strategy $x \in \mathcal{X}_i$, the contribution of player $i$ to the probability of the game ending in $z$ is computed as

$$\rho_i^x(z) = \sum_{\sigma \in \Sigma_i^*(z)} x_\sigma,$$

which is linear in $x$.

Now, we show that $\Omega_i^{\Gamma}$ is a convex polytope. By definition, $\Omega_i^{\Gamma} = f_i(\mathcal{X}_i)$ is the image of a convex polytope under a linear function, and therefore it is a convex polytope itself. □

**Theorem 2.** *Consider a game $\Gamma$. If player $i$ is not absent-minded, then $\Omega_i^{\Gamma} = \mathrm{co}\big(\tilde{\Omega}_i^{\Gamma}\big)$.*

*Proof.*

($\subseteq$) We know as a direct consequence of Lemma 1 that $\Omega_i^{\Gamma} = \mathrm{co}\{f_i(\sigma) : \sigma \in \Sigma_i\}$. Since every pure normal-form plan is also a behavioral strategy, $f_i(\sigma) \in \tilde{\Omega}_i^{\Gamma}$ for all $\sigma \in \Sigma_i$. Hence, $\Omega_i^{\Gamma} = \mathrm{co}\{f_i(\sigma) : \sigma \in \Sigma_i\} \subseteq \mathrm{co}(\tilde{\Omega}_i^{\Gamma})$.

($\supseteq$) Finally, we prove that that $\Omega_i^{\Gamma} \supseteq \mathrm{co}(\tilde{\Omega}_i^{\Gamma})$. Since $\Omega_i^{\Gamma}$ is convex, it is enough to show that $\Omega_i^{\Gamma} \supseteq \tilde{\Gamma}_i^{\Gamma}$. In other words, it is enough to prove that every behavioral-realization is also a realization in the sense of Definition 2, provided that player $i$ is not absent-minded. This is a well-known fact, and we refer the reader to Theorem 6.11 in the book by Maschler et al. [15].

□

**Lemma 3.** *For any $\Gamma$, $\Omega_{\mathcal{T}}^{\Gamma} = \mathrm{co}\left(\bigcup_{\sigma \in \Sigma_1} \Omega_{\mathcal{T}}^{\Gamma_\sigma}\right)$.*

*Proof.*

($\supseteq$) We start by proving that, for all $\sigma_1 \in \Sigma_1$, $\Omega_{\mathcal{T}}^{\Gamma_{\sigma_1}} \subseteq \Omega_{\mathcal{T}}^{\Gamma}$. Indeed, as a direct consequence of Lemma 1,

$$\begin{aligned}
\Omega_{\mathcal{T}}^{\Gamma_{\sigma_1}} &= \mathrm{co}\left(\{f_{\mathcal{T}}^{\Gamma}(\sigma_1, \sigma_2) : \sigma_2 \in \Sigma_2\}\right) \\
&\subseteq \mathrm{co}\left(\{f_{\mathcal{T}}^{\Gamma}(\sigma_1', \sigma_2) : \sigma_1' \in \Sigma_1, \sigma_2 \in \Sigma_2\}\right) \\
&= \Omega_{\mathcal{T}}^{\Gamma}.
\end{aligned}$$

Thus,

$$\bigcup_{\sigma_1 \in \Sigma_1} \Omega_{\mathcal{T}}^{\Gamma_{\sigma_1}} \subseteq \Omega_{\mathcal{T}}^{\Gamma}$$

and therefore, using the monotonicity of the convex hull function,

$$\mathrm{co}\left(\bigcup_{\sigma_1 \in \Sigma_1} \Omega_{\mathcal{T}}^{\Gamma_{\sigma_1}}\right) \subseteq \mathrm{co}(\Omega_{\mathcal{T}}^{\Gamma}) = \Omega_{\mathcal{T}}^{\Gamma},$$

where the last equality holds by convexity of $\Omega_{\mathcal{T}}^{\Gamma}$ (Lemma 1).

($\subseteq$) Take $\omega \in \Omega_{\mathcal{T}}^{\Gamma}$; we will show that $\omega \in \mathrm{co}\left(\bigcup_{\sigma \in \Sigma_1} \Omega_{\mathcal{T}}^{\Gamma_{\sigma}}\right)$ by exhibiting a convex combination of points in the polytopes $\{\Omega_{\mathcal{T}}^{\Gamma_{\sigma}} : \sigma \in \Sigma\}$ that equals $\omega$. By definition of realization function (Definition 1), $\omega$ is the image of a normal-form strategy $\alpha \in \Delta^{|\Sigma_1 \times \Sigma_2|}$ for the team. Hence, by linearity of the realization function $f_{\mathcal{T}}$ (Lemma 1),

$$\omega = \sum_{\substack{\sigma_1 \in \Sigma_1 \\ \sigma_2 \in \Sigma_2}} \alpha_{\sigma_1, \sigma_2} \, f_{\mathcal{T}}^{\Gamma}(\sigma_1, \sigma_2). \tag{1}$$

Now, define

$$\nu_{\sigma_1} := \sum_{\sigma_2 \in \Sigma_2} \alpha_{\sigma_1, \sigma_2}$$

for each $\sigma_1 \in \Sigma_1$. Clearly, each $\nu_{\sigma_1}$ is non-negative, and the sum of all $\nu_{\sigma_1}$'s is 1. Hence, from (1) we find that

$$\omega = \sum_{\substack{\sigma_1 \in \Sigma_1 \\ \nu_{\sigma_1} > 0}} \nu_{\sigma_1} \xi_{\sigma_1}, \quad \text{where} \quad \xi_{\sigma_1} := \sum_{\sigma_2 \in \Sigma_2} \frac{\alpha_{\sigma_1, \sigma_2}}{\nu_{\sigma_1}} f_{\mathcal{T}}^{\Gamma}(\sigma_1, \sigma_2).$$

Consequently, if we can show that $\xi_{\sigma_1} \in \Omega_{\mathcal{T}}^{\Gamma_{\sigma_1}}$ for all $\sigma_1 \in \Sigma_1 : \nu_{\sigma_1} > 0$, the proof is complete. Note that for all relevant $\sigma_1$, $\xi_{\sigma_1}$ is a convex combination of points in the set $\{f_{\mathcal{T}}^{\Gamma}(\sigma_1, \sigma_2) : \sigma_2 \in \Sigma_2\} \subseteq \Omega_{\mathcal{T}}^{\Gamma_{\sigma_1}}$. Finally, using the fact that $\Omega_{\mathcal{T}}^{\Gamma_{\sigma_1}}$ is convex (Lemma 1), we find $\xi_{\sigma_1} \in \Omega_{\mathcal{T}}^{\Gamma_{\sigma_1}}$, concluding the proof.

$\square$

**Corollary 1.** *The set of payoffs reachable in $\Gamma$ coincides with the set of payoffs reachable in $\Gamma^*$. Specifically, any strategy $\{\lambda_\sigma\}_{\sigma \in \Sigma_1}, \{\omega_\sigma\}_{\sigma \in \Sigma_1}$ over $\Gamma^*$ is payoff-equivalent to the realization-form strategy $\omega = \sum_{\sigma \in \Sigma_1} \lambda_\sigma \omega_\sigma$ in $\Gamma$.*

*Proof.* The payoff for the team in $\Gamma$ is equal to $\langle \sum_{\sigma \in \Sigma_1} \lambda_\sigma \omega_\sigma, y \rangle$, where $y \in \mathbb{R}^{|\cdot|}$ is a generic loss vector.

On the other hand, in $\Gamma^*$, $\mathcal{A}$ does not observe the initial move in $\Gamma^*$, and therefore the loss vector $y$ remains valid in each $\Gamma_\sigma$. Therefore, the team's payoff in $\Gamma^*$ is $\sum_{\sigma \in \Sigma_1} \lambda_\sigma \langle \omega_\sigma, y \rangle$. The two payoffs clearly coincide. $\square$

## C   Action sampling from $\Gamma^*$

Given a strategy profile $(\{\lambda_{\sigma_1}\}_{\sigma_1 \in \Sigma_1}, \{\omega_{\sigma_1}\}_{\sigma_1 \in \Sigma_1})$ over $\Gamma^*$, the goal is to draw a joint normal-form plan for the team (for the original game).

**Proposition 1.** *Letting $\omega = \sum_{\sigma_1 \in \Sigma_1} \lambda_{\sigma_1} \omega_{\sigma_1}$ and $\xi_{\omega_{\sigma_1}} \in f^{-1}(\omega_{\sigma_1})$, then*

$$\xi_\omega \triangleq \sum_{\sigma_1 \in \Sigma_1} \lambda_{\sigma_1} \xi_{\omega_{\sigma_1}} \in f^{-1}(\omega).$$

*Proof.* $f(\xi_\omega) = \sum_{\sigma_1 \in \Sigma_1} \lambda_{\sigma_1} f(\xi_{\omega_{\sigma_1}})$ by linearity. This is equal to $\sum_{\sigma_1 \in \Sigma_1} \lambda_{\sigma_1} \omega_{\sigma_1} = \omega$. □

The immediate way of sampling a joint normal-form action of $\Gamma$ from a realization $\omega$ over $\Gamma^*$ is the following. First, compute the set of joint actions required to form a normal-form strategy equivalent to $\omega$ (it is enough to adopt a simple greedy algorithm). Then, *recommend* each plan $(\sigma_1, \sigma_2)$ with probability $\xi_\omega(\sigma_1, \sigma_2)$ since $\xi_\omega \triangleq f^{-1}(\omega) \in \Delta^{|\Sigma_1 \times \Sigma_2|}$.

**Alternative method**. For Lemma 1, the normal-form plan $(\sigma_1, \sigma_2)$ is played with probability $\lambda_{\sigma_1} \xi_{\omega_{\sigma_1}}(\sigma_2)$. Therefore, we can first sample an action for Player 1 according to $\lambda_{\sigma_1}$. Then, we are left with the problem of finding $\xi_{\omega_{\sigma_1}} \in f^{-1}(\omega_{\sigma_1})$. It is enough to sample an element from $\mathbf{1}_{\sigma_1} \times \xi_{2,\omega_{\sigma_1}}$, where $\mathbf{1}_{\sigma_1} \in \Delta^{|\Sigma_1|}$ and $\xi_{2,\omega_{\sigma_1}} \in \Delta^{|\Sigma_2|}$.

## D   TMECor as a hybrid linear programming formulation

This section reviews prior techniques to compute *team-maxmin* equilibria with coordination devices. The leading paradigm to compute a TMECor is the *Hybrid Column Generation* algorithm introduced by Celli and Gatti [6]. This technique makes use of *hybrid* linear programs, which are based on the idea of letting team members play a joint normal-form strategy while the adversary still employs the sequence form. The idea of the algorithm is to proceed in a classical column generation fashion (see, e.g., [16]), generating progressively the set of joint normal-form plans of the team. The rationale is that there exists at least one TMECor with at most $|Q_\mathcal{A}|$ joint normal-form plans played with strictly positive probability by the team.

Consider $\mathcal{P} = \{1, 2, \mathcal{A}\}$, where $\mathcal{A}$ denotes the adversary (opponent) of the team. The algorithm progressively adds joint normal-form plans from $\Sigma_1 \times \Sigma_2$ to the set the set $\Sigma_{1\times 2}^{\text{cur}}$. A hybrid utility matrix $U_h$ is built along with $\Sigma_{1\times 2}^{\text{cur}}$. For each $\sigma_{1\times 2} \in \Sigma_{1\times 2}^{\text{cur}}$ a $Q_\mathcal{A}$-dimensional column vector is added to $U_h$. At each iteration of the algorithm, a *hybrid-maxmin* and a *hybrid-minmax* are employed to compute the equilibrium strategy profile for the current $U_h$. The *hybrid-maxmin* problem has $|Q_\mathcal{A}| + 1$ constraints and $|\Sigma_{1\times 2}^{\text{cur}}| + |\mathcal{I}_\mathcal{A}|$ variables, the *hybrid-minmax* is obtained by strong duality. Then, a new joint normal-form plan of the team is selected trough a best response oracle. These steps are iterated until the oracle returns a best response that is already contained in $\Sigma_{1\times 2}^{\text{cur}}$.

The problem of finding a joint normal-form plan of the team in best response to a given sequence-form strategy of the opponent is shown to be APX-hard (i.e., it does not admit a PTAS). The oracle of [6] employs a binary variable for each terminal node of the game. It produces two pure sequence-form strategies for the team members by forcing, for each $z \in Z$, the corresponding binary variable to be equal to 1 iff all team's sequences on the path to $z$ are selected with probability 1.

The main concern with this approach is that, with the growth of $\Sigma_{1\times 2}^{\text{cur}}$, LP's computations easily become an infeasible computational burden as the hybrid representation is exponential in the number of information sets of team members ($2^{|\mathcal{I}_1| + |\mathcal{I}_2|}$).

## E   Team best-response subroutine

Our subroutine looks for a pair $(\sigma^t, \omega_\mathcal{T}^t)$, with $\sigma \in \Sigma_1$, and $\omega_\mathcal{T}^t \in \Omega_\mathcal{T}^{\Gamma_\sigma}$, in best-response to a given $\bar{\omega}_\mathcal{A}$. In order to compute $(\sigma^t, \omega_\mathcal{T}^t)$, we employ the sequence-form strategies of the team defined over $\Gamma$. Specifically, the pure sequence-form strategy $r_1$ corresponds to selecting a $\sigma \in \Sigma_1$ at $\phi$ in $\Gamma^*$. Determining the (potentially mixed) sequence-form strategy for the other team member ($r_2$) is equivalent to computing $\omega_\mathcal{T}^t$ in the subtree selected by $r_1$.

Without loss of generality, we assume all payoffs of the team to be non-negative; indeed, payoffs can always be shifted by a constant without affecting the BR problem. In the following, sequence form constraints (see Section 2) are written, as customary, in matrix form as $F_i r_i = f_i$, where $F_i$ is an appropriate $|H_i| \times |Q_i|$ matrix and $f_i^\top = (1, 0, \ldots, 0)$ is a vector of dimension $|H_i|$.

The $\mathrm{BR}_{\mathcal{T}}(\bar{\omega}_{\mathcal{A}})$ subroutine consists of the following MILP:

$$\underset{w, r_1, r_2}{\arg\max} \sum_{q_1 \in Q_1} w(q_1) \tag{2}$$

$$\text{s.t. } w(q_1) \leq \sum_{q_2 \in Q_2} u_{q_1, q_2}^{\bar{\omega}_{\mathcal{A}}} r_2(q_2) \qquad \forall q_1 \in Q_1 \tag{3}$$

$$w(q_1) \leq M r_1(q_1) \qquad \forall q_1 \in Q_1 \tag{4}$$

$$F_1 r_1 = f_1 \tag{5}$$

$$F_2 r_2 = f_2 \tag{6}$$

$$r_2(q_2) \geq 0 \qquad \forall q_2 \in Q_2 \tag{7}$$

$$r_1 \in \{0, 1\}^{|Q_1|} \tag{8}$$

where $u^{\bar{\omega}_{\mathcal{A}}}$ is the $|Q_1| \times |Q_2|$ utility matrix of the team obtained by marginalizing with respect to the given realization of the opponent $\bar{\omega}_{\mathcal{A}}$. $r_1$ is a $|Q_1|$-dimensional vector of binary variables. The formulation can be derived starting from the problem of maximizing $r_1^\top u r_2$ under constraints (5)–(8). Let $a_{q_1} \triangleq \sum_{q_2} u_{q_1, q_2}^{\bar{\omega}_{\mathcal{A}}} r_2(q_2)$, and $w(q_1) \triangleq r_1(q_1) a_{q_1}$. Then, the objective function becomes $\sum_{q_1 \in Q_1} w(q_1)$. In order to ensure that, whenever $r_1(q_1) = 0$, $w(q_1) = 0$, the following constraints are necessary: $w(q_1) \leq M r_1(q_1)$ and $w(q_1) \geq 0$, where $M$ is the maximum payoff of the team. Moreover, in order to ensure that $w(q_1) = a_{q_1}$ holds whenever $r_1(q_1) = 1$, we introduce $w(q_1) \leq a_{q_1}$ and $w(q_1) \geq a_{q_1} - M(1 - r_1(q_1))$. It is enough to enforce upper bounds on $w$'s values (Constraints (3) and (4)) because of the objective function that we are maximizing and since we assume a positive utility for each terminal node.

In settings with more than two team members, our formulation enables one to pick any one team player's strategy and represent it using continuous variables instead of having binary variables for her in the best-response oracle MILP.

## F  Experimental evaluation

### F.1  Kuhn3-$k$

In Kuhn3-$k$ there are three players and $k$ possible cards. Each player initially pays one chip to the pot, and is dealt a single private card. Then, players act in turns. The first player may check or bet—put one additional chip in the pot. The second player either decides whether to check or bet after first player's check, or whether to fold/call the bet. If no bet was previously made, the third player decides between checking or betting. Otherwise, she has to fold or call. If the second player bet, the first player still has to decide between fold/call. If the third player bet, then both the first and the second have to choose between folding or calling the bet. At the showdown, the player with the highest card who has not folded wins all the chips in the pot.

### F.2  Leduc3-$k$

Leduc hold'em poker [21] is a widely-used benchmark in the imperfect-information game-solving community. In order to better evaluate the scalability of our technique, we employ a larger three-player variant of the game. In our enlarged variant, the deck contains three suits and $k \geq 3$ card ranks, that is, it consists of triples of cards $1, \ldots, k$ for a total of $3k$ cards.

Each player initially pays one chip to the pot, and is dealt a single private card. After a first round of betting (with betting parameter $p = 2$, see below), a community card is dealt face up. Then, a second round of betting is played (with betting parameter $p = 4$, see below). Finally, a showdown occurs and players that did not fold reveal their private cards. If a player pairs her card with the community card, she wins the pot. Otherwise, the player with the highest private card wins. In the event that all players have the same private card, they draw and split the pot.

Each round of betting with betting parameter $p$ goes as follows:

(1) Player 1 can check or bet $p$. If Player 1 checks, the betting round continues with Step (2); otherwise, the betting round continues with Step (8).

(2) Player 2 can check or bet $p$. If Player 2 checks, the betting round continues with Step (3); otherwise, the betting round continues with Step (6).

(3) Player 3 can check or bet $p$. If Player 3 checks, the betting round ends; otherwise, the betting round continues with Step (4).

(4) Player 1 can fold or call. If Player 1 folds, the betting round continues with Step (5); otherwise, Player 1 adds $p$ to the pot and the betting round continues with Step (5).

(5) Player 2 can fold or call. In either case the betting round ends. If Player 2 calls, she adds $p$ to the pot.

(6) Player 3 can fold or call. If Player 3 folds, the betting round continues with Step (7); otherwise, Player 3 adds $p$ to the pot and the betting round continues with Step (7).

(7) Player 1 can fold or call. If Player 1 calls, she adds $p$ to the pot. After Player 1's choice the betting round ends.

(8) Player 2 can fold or call. If Player 2 folds, the betting round continues with Step (9); otherwise, Player 2 adds $p$ to the pot and the betting round continues with Step (9).

(9) Player 3 can fold or call. If Player 3 calls, she adds $p$ to the pot. The betting round terminates after her choice.

## F.3  Worst case team utility

Table 4 shows the utility that the team is guaranteed to achieve in each game instance, with varying position of the opponent. These values are the worst case utilities, obtained when the opponent is best responding against the average team strategy. Specifically, let $\bar{\omega}'_{\mathcal{T}} \in \Omega^{\Gamma}_{\mathcal{T}}$ be the team realization over $\Gamma$ induced by the average team strategy $(\bar{\lambda}, (\bar{\omega}_{\mathcal{T},\sigma})_{\sigma \in \Sigma_1})$ (computed through Algorithm 1), and let $\omega^*_{\mathcal{A}} = \mathsf{BR}_{\mathcal{A}}(\bar{\lambda}, (\bar{\omega}_{\mathcal{T},\sigma})_{\sigma \in \Sigma_1})$. Then, the values are computed as, $\bar{\omega}'^{\top}_{\mathcal{T}} U \omega^*_{\mathcal{A}}$, where $U$ is a suitably defined (diagonal) $|Z| \times |Z|$ *payoff* matrix.

| Game | Team Utility | | |
|---|---|---|---|
| | Adv 1 | Adv 2 | Adv 3 |
| K3 | -0.0002 | -0.0002 | -0.0001 |
| K4 | 0.0369 | 0.0215 | -0.0474 |
| K5 | 0.0405 | 0.0137 | -0.0274 |
| K6 | 0.0499 | 0.0262 | -0.0267 |
| K7 | 0.0569 | 0.0271 | -0.0254 |
| L3 | 0.1533 | 0.0529 | -0.0412 |
| L4 | 0.0829 | -0.029 | -0.1901 |

Table 4: Worst case utilities for the team.

## F.4  Comparison between *ex ante* coordinated strategies and behavioral strategies

The results obtained on parametric Kuhn game instances with fictitious team-play are compared with the results obtained by computing the *team-maxmin* equilibrium [25], which is the best NE attainable without *ex ante* coordination of team members.

We employ fictitious team-play with 5000 iterations and a time limit of 15 seconds on the oracles' compute times (see also Table 2). A team-maxmin equilibrium is computed by solving a non-linear, non-convex optimization problem [6]. We employ AMPL 20181005, with the global optimization solver BARON 18.8.23 [22], and we set a time threshold of 15 hours.

Table 5 describes the results obtained in games where the opponent plays as the second player. Column *team-maxmin* displays the utility obtained when the opponent best-responds to the incumbent team strategies computed by the solver (BARON never reaches an optimal solution within the time limit).

| Game | team-maxmin | TMECor |
|---|---|---|
| K3 | $-6.03 \cdot 10^{-8}$ | 0.0004 |
| K4 | 0.0237 | 0.0335 |
| K5 | 0.0116 | 0.0205 |
| K6 | 0.0207 | 0.0329 |
| K7 | 0.0198 | 0.0333 |

Table 5: Comparison between the utility of the team at the *team-maxmin* equilibrium and at the TMECor.

*Ex ante* coordination always makes team members better off with respect to playing behavioral strategies.