[Reviews · NeurIPS 2018]

Reviewer 1



Summary: This paper explores the problem of finding the equilibrium of a game between an adversary and a team of players who can coordinate on strategies before the game starts. The authors first argue that this game can be expressed as a two-player game with imperfect recall (intuitively, we can treat the team as a meta-player since they can try to agree on some strategy beforehand). They then propose a new game representation, realization form, and introduce an auxiliary game that is easier to deal with (two-player with perfect recall) and is equivalent to the original game (in the sense that there exist mappings between strategy profiles in two games). The authors then modified the standard fictitious play method that takes advantage of the new structure and is able to find the equilibrium. Experiments are also conducted to demonstrate that the new presentation improves the efficiency in equilibrium computation. Comments: Overall, I think the idea of the proposed algorithm is interesting: utilize the property of the auxiliary game to guarantee fictitious play would converge to equilibrium, and use the original game to help efficiently find the best response in each iteration. I would lean towards accepting the paper. Since the contribution of the paper is a more efficient way of finding the equilibrium, it might be helpful if the authors can provide discussion on the efficiency issue? For example, the current paper has focused the discussion on the 2-player vs. the adversary case. The authors do explain that there is a straightforward way to extend it to more players, as mentioned in the conclusion. However, the size of the auxiliary game seems to grow significantly when the number of players increases. How much will this impact the performance of the equilibrium computation? The proposed algorithm relies on solving integer programs at each iteration. This seems to limit the scalability of the algorithm? In particular, I am wondering how much does it impact the algorithm when the number of pure strategies and/or the number of players increases? It seems to me the paper might be more suitable for AI or game theory conferences since the main contribution is in game representations and the discussion on learning is limited.

Reviewer 2



The paper presents a new representation of strategies, a number of theoretical results on the properties in team games, and an algorithm to compute a team max-min correlated equilibrium (TMECor). Results are given on 3-player parameterized versions of common Poker benchmark games (where the team game is interpreted to be two players colluding against the third.) This is a nice paper with some novel and significant contributions, but I am confused about a number of things. My assessment depends on clarifications needed by the authors, so I encourage them to answer the questions below in the rebuttal. However, apart from these points, the paper is well-written and conveys some complex ideas well. The necessary background knowledge in extensive-form games is rather steep, but the paper does a good mix of explaining the ideas on a high-level complemented by technical depth. The problem is important and there is little research on how to find a TMECor, so a new algorithm in this setting is welcome. This algorithm, in particular, has the anytime property which is particularly appealing. Questions: 1. In Definition 1, rho^x(z) where x is a normal-form strategy was never defined anywhere, or I missed it. Is it the sum of the rho^sigma(z) over all sigma(I) that x is mixing over times the probability x(sigma)? 2. Section 3, why are the same constraints of summing to 1 not included for wl? 3a. It is not especially clear what role the realization strategies play in this paper. It seems they are mostly used to prove the realization equivalences in the theory, but it seems strategies are never actually stored in this representation. Is that correct? 3b. It seems like there is a many-to-one mapping from normal-form strategies to realization-form strategies, in the sense that there can be many normal form strategies that induce the same realization form strategy. Is this true? So given a realization form strategy, one cannot find a unique normal form strategy in general? Isn't this a problem if the algorithms are to use this representation? 3c. The text claims that the realization form allows using the behavior strategies directly. But in section 5 the algorithm is defined over the sequence form (not behavior stratgies). Am I missing something? I found Section 5 difficult to understand. In particular, I am having trouble understanding how the theory developed in Sections 3-4 is leveraged by the algorithm. 4. What is the gradient \bar{g}_A? This is not a common component of ficitious play, and it is not defined anywhere. 5. Line 14: what does it mean to 'compute gradient by fixing r_1^t, r_2^t'? r_1^t, r_2^t are passed in as arguments; why would they not be "fixed", and what does this line do? (What is \hat{g}_A)? 6. The algorithm works on sequence-form strategies. Does this not directly conflict with the motivation that the team game can be thought of as an imperfect recall game? Is it that \Gamma^* is a perfect recall equivalent of the imperfect recall \Gamma? 7. It is unclear what parts of the algorithm operate within \Gamma^* and what parts in \Gamma. Can you clarify this? Other comments: - Page 1: "This is not the case for games with multiple players". Two players counts as "multiple players", so this should be rephrased to "with more than two players". - Page 4: Last sentence of Def 1, rho_i^{x_1} should be rho_i^{x_i} - Page 6: Lemma 2 uses a non-standard co(set) to mean that the left hand side is a convex combination of the set. Define this or simply write out ConvexCombination(set). Also the formal equals is not technically correct, why not just write it out: \Omega_T^\Gamma is a convex combination of set ? - Page 6: "tre" -> "tree" ************************** Post-rebuttal: The authors have clarified many points, thank you. One last point: I have not encountered the concept of a gradient (as defined by the authors) in fictitious play. I do not think it is "standard" (nor is its definition): a reference (and definition) should be provided.

Reviewer 3



The paper studies the problem of multi-player extensive-form games where players form teams. Within a team, players share the common payoff and the paper focuses on the zero-sum setting in the sense that two teams of players are playing against each other and the sum of utilities of both teams equals zero for each terminal state. Solving such games is challenging since they can be either modeled as multi player games with perfect recall, or as a two player game but with imperfect recall. The authors argue that within such setting, it is reasonable to consider solution concepts where the players of a team are able to correlate their strategies -- i.e., to agree in a advance on a probability distribution from which they sample their pure strategies. Under this assumption, the authors show that the range of expected utilities can be represented as a convex set and an auxiliary perfect recall game of an exponential size can be constructed, solving which leads to the solution of the original game. The authors propose a novel algorithm based on Fictitious Play in extensive-form games and experimentally demonstrate the performance on Kuhn and Leduc poker games with varying number of cards. Overall, the idea and the technical results seem interesting, however, I had quite some difficulties to understand why the realization form is not a straightforward consequence of the choice of the authors focus on normal-form strategies -- I would suggest to include a more complex example instead of Figure 1, namely one, where restricting to normal-form strategies does not directly lead to a perfect recall game. There are two concerns about the paper. First, it is difficult to agree with the statement that correlated strategies are the right choice for team games, especially due to the necessity of correlating before each game (or sampling the correlated strategies several times in advance, once for each play of 1 game). On the other hand, if you assume correlation via some actions during the play, then it is assumed that players have observable actions which limits the applicability (and can actually simplify the technical results). Such a correlation can make sense in other multi-agent adversarial scenarios (a team is fighting adversarial team). However, even in these tactical scenarios, the correlation with respect to receiving signals in each information set (instead of receiving one signal) seems more natural as argued by von Stengel and Forges when introducing Extensive-Form Correlated Equilibrium. Could the authors comment of that? Is it truly reasonable to focus on computing correlation over pure strategies in these games? And if so, do authors actually compute a correlated equilibrium in these games? How does TMECor differ from standard correlated equilibrium (not EFCE) in these games? Second, the authors omit the recent algorithms for computing behavioral strategies in imperfect recall games (namely works by Cermak et al. "Combining Incremental Strategy Generation and Branch and Bound Search for Computing Maxmin Strategies in Imperfect Recall Games" and "Approximating maxmin strategies in imperfect recall games using A-loss recall property"). These works are computing optimal strategies in behavioral strategies in imperfect recall games -- that, as the authors state, leads to a non-linear, non-convex optimization problem. At the same time, the scalability of the existing algorithms reported in these prior works outperforms the running time of the algorithm in this paper by several orders of magnitude (number of sequences is 10^6 and higher solvable within hours in the works by Cermak et al., compared to 10^2 in this submission). Since one of the positive results of this submission is to simplify the strategy space and offer better and more scalable algorithms, it seems that current experimental results do not strongly support this claim. A natural question arises -- can the techniques and algorithms for computing behavioral strategies in imperfect recall developed by Cermak et al. be adopted for computing the solution concept the authors propose (technically, it is a bilinear mathematical program, so it actually can be simplified)? If not, why? After rebuttal: I am not that satisfied with the response of the authors since several questions were left unanswered. 1. I am aware that ex-ante correlation does not lead to a perfect recall game. I was merely suggesting the authors to use a more complex example in the paper so that it is clearly stated in the paper. 2. I am not convinced that ex-ante correlation is the right choice, however, this my subjective opinion and I admit that ex-ante correlation has its place in the literature/research. The authors did not respond to my question regarding TMECor -- it seems to me that TMECor is a combination of two standard correlated equilibria played within the members of a team, joined in a zero-sum competition. If this is true, I am wondering whether there is some connection to the property that the realization form is a convex polyhedron -- e.g., does this mean that any CE in an imperfect recall game can be represented using the same strategy space? 3. The problem with scalability is as follows -- the authors suggest that they use TME for conceptual simplicity that should be positively reflected in the scalability. However, this is not currently the case, since the cited papers are able to solve larger games in behavioral strategies (that have all the negative aspects the authors state -- non-existence of Nash equilibria, non-linearity, non-convexity, etc.) compared to the current experiments in the submission. My question was, whether the advanced techniques that lead to this better scalability in the more complex strategy space can be modified to the proposed solution concept (i.e., not to directly apply cited algorithms -- which is what the authors responded to). Overall, there are some concerns over the significance of the paper (connections to the standard solution concepts, scalability), however, there is some contribution and I can imagine that the paper can be helpful for other researchers in a follow-up work.